# Effect of Different Irrigation Solutions on the Diffusion of MTA Cement into the Root Canal Dentin

**DOI:** 10.3390/ma13235472

**Published:** 2020-12-01

**Authors:** José Pedro Martinho, Sara França, Siri Paulo, Anabela Baptista Paula, Ana Sofia Coelho, Ana Margarida Abrantes, Francisco Caramelo, Eunice Carrilho, Carlos Miguel Marto, Maria Filomena Botelho, Henrique Girão, Manuel Marques-Ferreira

**Affiliations:** 1Institute of Endodontics, Faculty of Medicine, University of Coimbra, 3000-075 Coimbra, Portugal; josepedromartinho@gmail.com (J.P.M.); sarafranca@gmail.com (S.F.); sirivpaulo@gmail.com (S.P.); 2Coimbra Institute for Clinical and Biomedical Research (iCBR), Area of Environment, Genetics and Oncobiology (CIMAGO), University of Coimbra, 3000-548 Coimbra, Portugal; anabelabppaula@sapo.pt (A.B.P.); anasofiacoelho@gmail.com (A.S.C.); mabrantes@fmed.uc.pt (A.M.A.); eunicecarrilho@gmail.com (E.C.); cmiguel.marto@uc.pt (C.M.M.); mfbotelho@fmed.uc.pt (M.F.B.); 3Center for Innovative Biomedicine and Biotechnology (CIBB), University of Coimbra, 3000-548 Coimbra, Portugal; fcaramelo@fmed.uc.pt (F.C.); hmgirao@fmed.uc.pt (H.G.); 4Clinical Academic Center of Coimbra, CACC, 3004-561 Coimbra, Portugal; 5Institute of Integrated Clinical Practice, Faculty of Medicine, University of Coimbra, 3000-075 Coimbra, Portugal; 6Institute of Biophysics, Faculty of Medicine, University of Coimbra, 3000-548 Coimbra, Portugal; 7Laboratory of Biostatistics and Medical Informatics, Faculty of Medicine, University of Coimbra, 3000-548 Coimbra, Portugal; 8Coimbra Institute for Clinical and Biomedical Research (iCBR), University of Coimbra, 3000-548 Coimbra, Portugal; 9Institute of Experimental Pathology, Faculty of Medicine, University of Coimbra, 3000-548 Coimbra, Portugal

**Keywords:** bioceramics, calcium silicate sealer, chlorhexidine, endodontic sealers, mineral trioxide aggregate, Rhodamine B, tricalcium phosphate

## Abstract

(1) Aim: This study aims to analyze the in vitro infiltration of a silicate root canal sealer into dentinal tubules after using different endodontic irrigating solutions. (2) Methods: Twenty-nine teeth with single roots were separated into three groups according to the final irrigation protocol: G1 *n* = 10) = 17% EDTA (ethylenediaminetetraacetic acid) + 3.0% sodium hypochlorite (NaOCl), G2 (*n* = 10) = 17% EDTA + 2.0% chlorhexidine and G3 (Control group, *n* = 9) = 17% EDTA + saline solution. Root canals were filled using cold lateral compaction technique with MTA Fillapex sealer and gutta-percha. The sealer was labeled with rhodamine B. The teeth were segmented at the middle and third apical sections, which were visualized using 10× confocal laser microscopy to determine the sealer penetration percentage. (3) Results: In the apical section, no statistically significant differences were found between the groups regarding sealer penetration. In the middle section, Group 1 obtained the highest percentage, and Group 2 the lowest (*p* = 0.004). Group 1 also presented statistically significant differences in the Control Group (*p* = 0.031) and had close sealer penetration values. Meanwhile, the Control Group (*p* = 0.023) and Group 2 (*p* = 0.029) revealed a significant decrease of sealer penetration between the apical and middle sections. (4) Conclusion: The obtained results support that final irrigation with NaOCl promoted similar sealer penetration in the apical and middle sections. On the other hand, a significant decrease in the sealer penetration of the middle section was observed for the chlorhexidine and saline groups. Compared to other irrigant solutions, NaOCl promotes more uniform sealer penetration, which can correlate with better sealing and, consequently, higher endodontic treatment success.

## 1. Introduction

Endodontic treatment aims to maintain healthy periapical tissue and prevent the reinfection of the root canal system. For this, it is necessary to eliminate inflamed or necrotic pulp tissue, bacteria and their byproducts through chemo-mechanical preparation of the root canal [1,2].

Mechanical elimination may have a limited effect, since the instruments may not reach and remove the root canal infected tissues [3,4,5]. In addition to a substantial part of the root walls not being touched by these instruments [6], mechanical preparation promotes the formation of an organic and inorganic debris layer designated by the smear layer [7]. The presence of this 2–5 µm thick layer prevents the penetration of the intracanal medication and the filling material into the dentinal tubules, so its removal is vital to improving the adaptation of the filling material to the root canal walls [7,8].

Therefore, it is necessary to complement the mechanical instrumentation with the use of chemical irrigants [1], such as sodium hypochlorite (NaOCl), chlorhexidine gluconate (CHX) and ethylenediaminetetraacetic acid (EDTA). These irrigants are the most widely used and generally remove the organic tissue and debris, as well as the pathogenic microorganisms and the smear layer, cleaning the dentin surface [3,4].

Sodium hypochlorite is the most widely used solution [3]. Although NaOCl has a strong antimicrobial action and an excellent tissue-dissolving capacity, this irrigant cannot dissolve the inorganic component of the smear layer, and is a strong irritant to periapical tissues, even at concentrations lower than 0.1%, mainly due to its basic pH of around 11 [3,4,5,9]. Its oxidizing activities produce an oxygen-rich layer that decreases resistance and increases the microleakage in adhesive resins [1,3,10].

Chlorhexidine gluconate, a cationic bisbiguanide, is an antimicrobial agent with good substantivity activity [5,9]. This solution has broad-spectrum antimicrobial activity, being active against gram-negative and -positive bacteria, but does not dissolve organic tissue or the inorganic component of the smear layer [5,11,12,13]. For this reason, it is advisable to use a chelating agent such as EDTA to remove the inorganic component of the smear layer and demineralize dentin. This polyprotic acid is clinically used in concentrations between 10% and 17%, and decalcifies dentin to a depth of 20–30 µm in 5 min [1,8,14]. Although all irrigation solutions have excellent properties, none can dissolve residual organic tissue and eliminate the smear layer simultaneously [1].

In addition to the chemo-mechanical preparation, a successful root canal treatment includes the three-dimensional filling of the root canal system with a sealer penetration into the dentinal tubules [3,15]. This penetration allows entombing and prevents bacterial invasion, forming a bond between the root canal walls and the core of the filling material [16,17].

According to their chemical constituents, endodontic sealers are used in conjunction with core filling materials and can be classified as calcium hydroxide, zinc oxide eugenol, glass ionomer, resin, silicone and, more recently, bioceramic-based sealers [16]. Epoxy resin-based sealers are considered the gold standard due to their biological, physical and chemical properties and sealing capacity [18]. Recently, bioceramic-based sealers have emerged containing calcium phosphate, with a crystalline structure and a chemical composition similar to bone and tooth apatite materials. In addition to their excellent sealing capacity and mechanical and physical properties, their biocompatibility prevents rejection by the surrounding tissue [16,19]. A recent systematic review stated that the penetration of bioceramic cement is substantially more significant than that of epoxy resin-based sealers (AH Plus^®^, for example), even with activated irrigation methods and chelating agents [20]. Several root canal filling techniques fill the irregularities and voids in the root canal system, but the most widely used in clinical practice are lateral compaction, warm vertical compaction and core-carrier techniques [7,21]. For better sealer penetration in the dentinal tubules, it is necessary to remove organic material and smear the layer from the root canals. Failure to remove this layer alters the sealer penetration and compromises a satisfactory seal [8].

Mineral trioxide aggregate (MTA) is an example of a bioceramic-based cement, composed of tricalcium oxide, silicate oxide and tricalcium silicate [22]. In addition to its basic pH and chemical stability, MTA is nontoxic, has excellent biocompatibility and can set in the presence of moisture and blood [22]. MTA Fillapex^®^ (Angelus Industria de Produtos Odontologicos S/A, Londrina, Brazil), an example of an MTA-based sealer presented in a paste/paste system, is composed of mineral trioxide aggregate, bismuth trioxide, nanoparticulated silica, pigments and salicylate, as well as natural resin [22,23,24,25,26]. According to the manufacturer, this sealer’s setting reaction is due to its expansion in the presence of moisture in the dentinal tubules [26]. This material also has excellent radiopacity, high solubility, good working time and easy handling [22,23,24,25,26].

This study aimed to evaluate the penetration depth of a calcium silicate-based sealer in the dentinal tubules after using different endodontic irrigating solutions. The null hypothesis was that there were no significant differences among the groups.

## 2. Materials and Methods

This study was approved by the Ethics Committee of the Faculty of Medicine of the University of Coimbra (117-CE-2017).

### 2.1. Sample Collection

In this in vitro study, 29 single-rooted extracted teeth with complete root formation were used. After extraction, the teeth were stored at 4 °C in a solution composed of 0.9% sodium chloride and 0.02% sodium azide for two weeks. Radiographs were taken from the facial and proximal views to ensure the presence of a single canal. Subsequently, the crowns were sectioned with a high-speed burr and water spray to obtain 15 mm long roots.

### 2.2. Root Canal Preparation

Preparation of the root canals was performed using ProTaper^®^ nickel-titanium rotary instruments (Dentsply Maillefer, Ballaigues, Switzerland), using a handpiece with an electric motor (X-Smart, Dentsply Maillefer, Ballaigues, Switzerland) at 250 rpm. The apical patency was verified with an ISO size K-file 10 (Dentsply Maillefer, Ballaigues, Switzerland). An F3 ProTaper^®^ was used to file the samples to the appropriate working length.

After preparation, the samples were irrigated using 3 mL of 3.0% NaOCl (CanalPro^®^, Coltène/Whaledent Inc., Cuyahoga Falls, OH, USA) using conventional irrigation with a 27-gauge endodontic needle (Kendall Monoject^®^, Tyco/Healthcare, Faridabad, India) adapted to a syringe and positioned 3 mm short of the working length. The roots were randomly divided into two experimental groups and one control group.

The same operator performed all root canal preparations.

### 2.3. Final Irrigation Protocol

After root canal preparations were complete, all groups were irrigated for 1 min with 3 mL of 17% EDTA (Coltène/Whaledent Inc., Langenau, Germany, D-89122), followed by irrigation with NaOCl or CHX in the experimental groups—3 mL of 3.0% NaOCl for 3 min in Group 1 (G1-NaOCl, *n* = 10); 3 mL of 2.0% CHX (CanalPro, Coltène/Whaledent Inc., Langenau, Germany) for 3 min in Group 2 (G2-CHX, *n* = 10). This was followed by irrigation with saline solution in the G3 (Control group-SS, *n* = 9). After the final irrigation protocol, a final flush with 5 mL of saline solution to neutralize the solutions was used, and the canals were dried with paper points (Dentsply Maillefer, Ballaigues, Switzerland).

### 2.4. Root Canal Filling

For fluorescence analysis under confocal laser microscopy, fluorescent Rhodamine B dye (Panreac^®^, Barcelona, Spain) was mixed with MTA Fillapex^®^ sealer (Angelus, Londrina, PR, Brazil) to an approximate concentration of 0.1%. This mixture was placed in the canals with a master cone gutta-percha [23]. The root canals were filled with the master cone gutta-percha size #30 (Dentsply, Maillefer, Ballaigues, Switzerland) under a cold lateral condensation technique, with additional gutta-percha size #20 points. After removing the excess of gutta-percha at the canal open with a hot instrument, a #60 hand plugger (Dentsply, Maillefer, Ballaigues, Switzerland) was used with vertical pressure for final compaction. The root canal orifices were sealed with a temporary restoration, namely Cavit^®^ (3M, ESPE AG, Seefeld, Germany), and were saved at 100% relative humidity and 37 °C for two weeks. 

### 2.5. Sectioning and Image Analysis

One-millimeter-thick transverse sections were cut with a precision cutting machine (Exact 310 CP, Kulzer Exact, Hanau, Germany) at 5 and 10 mm to the apex, obtaining dentin discs of the middle and apical thirds of each root, resulting in six distinct groups (Table 1). All sections were sequentially polished under water cooling with diamond discs of decreasing granulometry (800, 1000, 2500, 4000) (Hermes, Hamburg, Germany) in a polishing unit with a rotating plate (Exact 400CS, Advanced Technologies GmbH, Norderstedt, Germany). 

After polishing, the sections were placed on glass slides for further image acquisition. The samples were observed and photographed using a laser scanning microscope (Zeiss 710, Carl Zeiss, Gottingen, Germany) at an excitation laser wavelength of 561 nm, and using the fluorescent mode with an EC-Plan-Neofluor 10×/0.3 M27 objective. Each image obtained at 10× had an area of 1414.22 × 1414.22 μm^2^ and a resolution of 512 × 512 pixels. 

Using Adobe Photoshop^®^ 7.0 (Adobe Systems, San Jose, CA, USA), the different images from each sample were overlaid to obtain a single image for analysis. After this, the Image J^®^ software (Version 1.53, Rasband, W.S., ImageJ, U. S. National Institutes of Health, Bethesda, MD, USA) was used to outline and measure the regular fluorescent ring around the canal wall for each sample (Figure 1 and Figure 2).

Figure 2 represents the measurement of the area along the canal wall in which the sealer had penetrated the dentinal tubules. The sealer penetration in each section was calculated using the formula: sealer penetration area divided by the canal circumferential area [27].

### 2.6. Statistical Analysis

The IBM SPSS software (Version 19, IB5M Corporation, Armonk, NY, USA) was used for the statistical analysis. A significance level of 5% was considered for all statistical tests, and statistical analyses were carried out independently in the two sections (middle and apical).

The obtained data were evaluated for normality using the Shapiro-Wilk test. For data with normal distribution, the Student’s *t*-test was used to compare quantitative variables between two groups, while comparisons of more than two groups were performed using one-factor ANOVA with post hoc analysis using Tukey’s test.

For data with not-normal distribution, nonparametric tests were used. The Mann-Whitney U test was used to compare the quantitative variables between the two groups, and the Kruskal-Wallis test, with multiple comparisons performed using the Mann-Whitney U test with Bonferroni correction, was used for comparisons of more than two groups.

## 3. Results

The greatest sealer tubule penetration was observed in the apical sections in the SS Group, followed by Group 1 (NaOCl) and Group 2 (CHX), although without a statistically significant difference (Table 1 and Figure 3).

Group 1 (NaOCl) obtained the highest percentage in the middle sections, and Group 2 (CHX) the lowest, with statistically significant differences (*p* = 0.004) between them. In the middle sections, Group 1 also presented statistically significant differences in the Control Group (SS) (*p* = 0.031) (Table 1, Figure 4 and Figure 5A).

Regarding regional variance for the same irrigant, NaOCl (Group1) had comparable values of sealer penetration in the two root sections evaluated. Group 2 obtained significant differences (*p* = 0.029) of sealer penetration between the apical and middle section, and the same was verified in the Control Group (*p* = 0.023) (Figure 5B).

## 4. Discussion

The present study aimed to compare the penetration of MTA Fillapex^®^ into dentinal tubules after final irrigation with saline solution, chlorhexidine and sodium hypochlorite.

Regarding the experimental design, the number of teeth was determined based on previous similar studies [2,3,9,24]. The obtained results presented statistically significant differences in some testing conditions, supporting the *n* in each group was sufficient. Comparison between the different root thirds (apical, middle and coronal) provided essential information regarding the different anatomical characteristics and material behavior in each area, as explored in several papers [28,29,30,31]. In this study, we focused on the apical and middle thirds, since the sealer penetration in these areas is fundamental to the success of endodontic treatment.

The use of confocal laser Scanning microscopy (CLSM) allowed us, in a nondestructive way and without requiring special specimen processing, to visualize the dentin-sealer interface and the penetration of the fluorescently labeled MTA Fillapex^®^ sealer [27,32]. Since we aimed to determine the penetration of the sealer into the root dentinal tubules, the incorporation of Rhodamine B into the sealer was essential to observe the extent of sealer adaptation and penetration, as proven previously [33,34]. Since the obturation technique with a silicate-based sealer does not influence the penetration of the sealer in the apical third of the root canal [26], the cold lateral condensation technique was used. Also, because this study intended to assess sealer penetration into dentinal tubules and not coronal microleakage, Cavit^®^ was used because of its sealing capacity, availability and ease of handling [34].

As previously noted, although no differences were observed in the apical section, both the CHX and SS groups promoted a significant decrease in sealer penetration in the middle section, which was not observed in the NaOCl group.

The Group 2 (CHX) results showed a lower penetration of the sealer in the middle third, even with 17% EDTA irrigation. CHX cannot dissolve organic tissue [9], while EDTA decalcifies dentin and leaves organic residues in the root walls, compromising the sealer’s adhesion to the canal walls. The latter may have influenced the obtained results due to insufficient EDTA use, a lack of agitation or insufficient contact time, keeping a dense insoluble precipitate—i.e., Para-chloroanaline (PCA)—in the dentinal tubules. This precipitate that forms between the NaOCl and the CHX coats the canal wall and causes the obstruction of the dentinal tubules along the root canal [5], causing a decrease in the sealer’s penetration. Through an SEM study, Akisue et al. (2010) concluded that there was a decrease in dentin permeability in the apical third due to the formation of a precipitate after combining 1% NaOCl and 2% CHX solutions [9]. This way, when CHX is used as an irrigant, larger volumes of EDTA and its activation should be used to improve the penetration of the sealer. Furthermore, the observed results may be due to the characteristics of the canal morphology, namely, the different densities and more significant variation of the dentinal tubules along the root canal [8].

On the other hand, when irrigated with NaOCl after EDTA, these remnants are removed and could help the sealer’s penetration. This occurs because the use of NaOCl after EDTA removes the organic matrix and increases the exposure of inorganic components, with demineralization and modification of collagen-rich dentin into a structure with multiple irregularities in the inter- and peri-tubular dentin [35]. This increase of irregularity can provide a larger area for the fluidity of MTA Fillapex^®^ to enhance penetration. Also, in a study by de Assis et al. (2010) where the contact angle between two endodontic sealers and the dentin treated with 5.25% NaOCl and 2% CHX was evaluated, it was observed that the contact angle in the absence of a smear layer after the use of EDTA showed lower values than when this chelant agent was not used [2].

This can also explain the decrease of sealer penetration in the saline solution group, where the lack of NaOCl after EDTA also maintained the organic remnants, similar to the CHX group, demonstrating the importance of final irrigation with NaOCl.

Kuçi et al. studied the penetration of AHPlus^®^ 26 and MTA Fillapex^®^ into the dentinal tubules of instrumented root canals that were obturated with cold or warm lateral compaction; they concluded that removing the smear layer increased the penetration of the MTA Fillapex^®^, and that there was greater penetration of this sealer when used with the cold lateral compaction technique [7]. Borges et al. studied the physicochemical properties of MTA Fillapex^®^, namely, the solubility, and concluded that it presented a higher mean value (2.88 ± 0.48), while AH Plus^®^ presented a lower mean value (0.56 ± 0.48) [15]. 

Chlorhexidine gluconate has antibacterial activity and the ability to adhere to hydroxyapatite, remaining active following root canal treatment, which justifies its use as an endodontic irrigant. However, as it cannot remove the smear layer, remnants of these organic materials may prevent the adhesion of the sealer to the root canal walls.

Although MTA-based sealers are hydrophilic, and the presence of water promotes their configuration, the degree of moisture does not affect their chemical components. It can still change the relative quantity of penetration and adaptation to the root canal dentin [2,3,23,26]. Even after drying the root canal with a paper point, as previously described in this study, the residual moisture, presented mainly in the apical third, seemed to favor sealer penetration [2,3,23,26].

Although the use of rhodamine B is widely described for evaluating sealer penetration, more precise methods can be used, such as calcium-affine marker Fluo-3. Instead of rhodamine B, Fluo-3 is a nonfluorescent compound that binds selectively to the calcium present in calcium silicate-based sealer. The presence of these ions increases fluorescence, decreasing false-positive results when compared to other dyes, namely rhodamine B [28,34].

Further studies using other techniques to evaluate sealer penetration should be performed to compare to the obtained results, clarifying the penetration of the sealer and the validity of the technique.

## 5. Conclusions

Within the limitations of the present study, it can be concluded that in the apical third, the percentage of sealer penetration demonstrated no significant differences between the different irrigants tested. In the middle section, only final irrigation with NaOCl was responsible for similar sealer penetration in the apical third. For the chlorhexidine and saline groups, a significant decrease in sealer penetration was observed.

This study supports the hypothesis that final irrigation with NaOCl increases the calcium silicate sealer penetration, which results in better sealing, and consequently, the possibility of a higher endodontic success rate.

## Figures and Tables

**Figure 1 materials-13-05472-f001:**
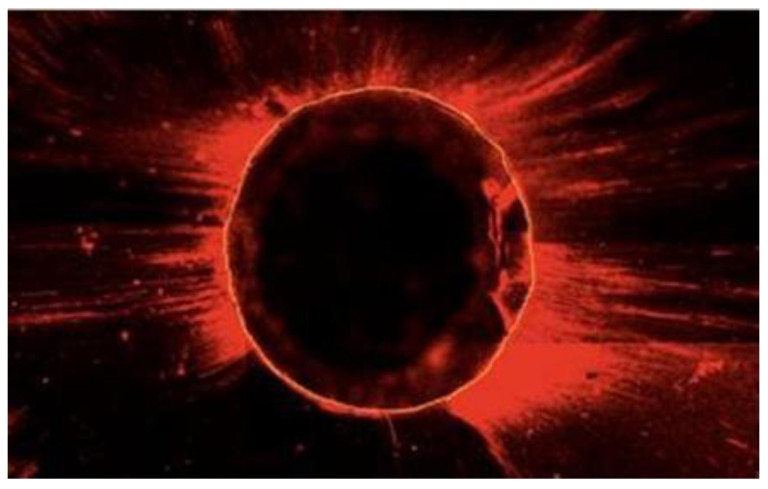
Image of the root canal section, where the circumference area was measured by Image J^®^.

**Figure 2 materials-13-05472-f002:**
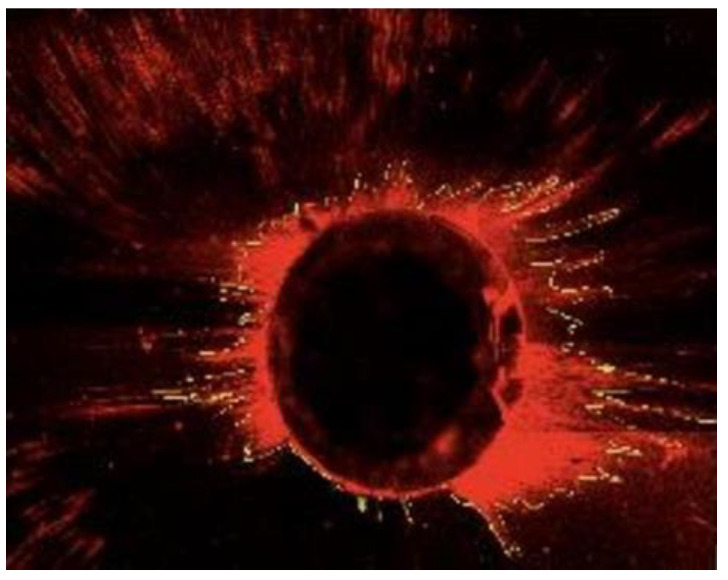
Area of sealer penetration into dentinal tubules measured by ImageJ^®^.

**Figure 3 materials-13-05472-f003:**
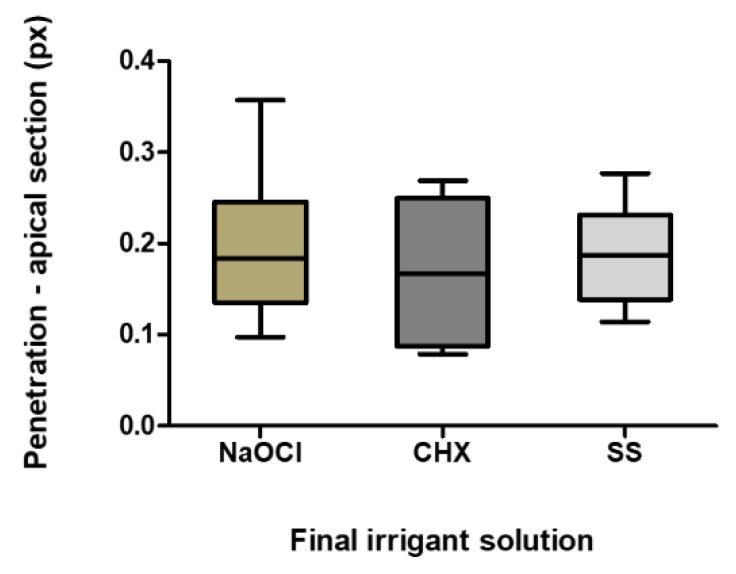
Sealer penetration in the apical sections (pixels). The boxplot shows the mean, minimum and maximum penetration of the different irrigants in the apical section.

**Figure 4 materials-13-05472-f004:**
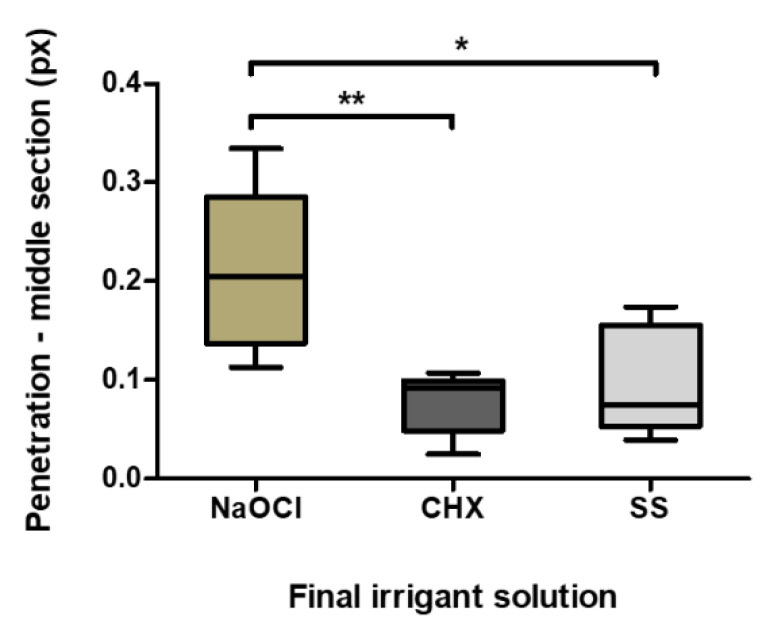
Sealer penetration in the middle sections (pixels). The boxplot shows the mean, minimum and maximum penetration of the different irrigants in the apical section. * *p* = 0.031; ** *p* = 0.004.

**Figure 5 materials-13-05472-f005:**
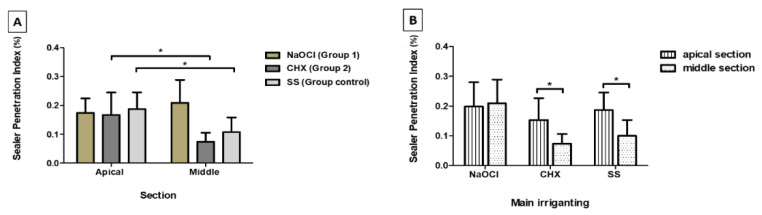
Mean percentage of sealer penetration in the two analyzed sections and in the different final irrigation protocols: **A**—Section analysis (pixels); * *p* < 0.05; **B**—Irrigant solution analysis (pixels); * *p* < 0.05.

**Table 1 materials-13-05472-t001:** Sealer penetration for the apical and middle section of the canals.

Section	Irrigant Solution	Mean	Median	Sd	Min	Max
Apical	Group 1 (NaOCl)	0.174	0.164	0.051	0.098	0.255
	Group 2 (CHX)	0.167	0.167	0.078	0.079	0.269
	Control Group (SS)	0.187	0.187	0.058	0.114	0.277
Middle	Group 1 (NaOCl)	0.209	0.205	0.080	0.113	0.334
	Group 2 (CHX)	0.074	0.092	0.032	0.025	0.107
	Control Group (SS)	0.108	0.102	0.051	0.046	0.174

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
