# Peer review of "Effect of Different Irrigation Solutions on the Diffusion of MTA Cement into the Root Canal Dentin"

_materials, 2020, doi:10.3390/ma13235472_

Round 1

Reviewer 1 Report

Thank you for giving me this opportunity to review the in vitro study article entitled, "Effect of final irrigation on silicate root canal sealer penetration into dentinal tubules".

I here carefully reviewed the submitted set of the manuscript and found it possibly merits of publication. However, there need substantial required to meet the standard of scientific publication. This article should be re-evaluated after major revision to re-consider for a possibility of publication.

  1. The authors must follow the author guidelines. There are so many technical and basic errors/flaws in sentences and structures in the manuscript style of Materials, which seems to be hardly understandable as a scientific article submission throughout the manuscript.  
  2. The authors used human teeth. They must submit the ethical review certification in IRB. This study must be accepted before conducting this research.
  3. The Discussion sections are almost the repetitions of the Results sections. The authors must revise the Discussion from the very beginning referring to the similar and recent research literature.

Author Response

Thank you for giving me this opportunity to review the in vitro study article entitled,
"Effect of final irrigation on silicate root canal sealer penetration into dentinal
tubules".
I here carefully reviewed the submitted set of the manuscript and found it possibly
merits of publication. However, there need substantial required to meet the standard
of scientific publication. This article should be re-evaluated after major revision to reconsider for a possibility of publication.
1. The authors must follow the author guidelines. There are so many technical
and basic errors/flaws in sentences and structures in the manuscript style of
Materials, which seems to be hardly understandable as a scientific article
submission throughout the manuscript.
Answer: We greatly appreciate your comment. The manuscript was corrected to
match the Materials style.
2. The authors used human teeth. They must submit the ethical review
certification in IRB. This study must be accepted before conducting this
research.
Answer: The reviewer is correct, as we did ask for ethical approval before the
study begin. The ethical approval was added to the manuscript at lines 127 and
128.
3. The Discussion sections are almost the repetitions of the Results sections. The
authors must revise the Discussion from the very beginning referring to the
similar and recent research literature.
Answer: The Discussion section was changed, in order to include and explain
similar methodologies that prove similar results to those found in this article.

Reviewer 2 Report

It is a pleasure to contribute as a reviewer to Materials After reviewing the manuscript materials-987145 - Effect of final irrigation on silicate root canal sealer penetration into dentinal tubules, my recommendation is to accept with minor revision.

The information within this manuscript is accurate and should have minor corrections on the grammar and language before it is accepted for publication. The study does bring substantial contribution to the field of study. All information is well discussed, and the results support conclusions.

My suggestion is to change the title to: Effect of different irrigation solutions on the diffusion of MTA cement into the root canal dentin.

Author Response

It is a pleasure to contribute as a reviewer to Materials After reviewing the manuscript
materials-987145 - Effect of final irrigation on silicate root canal sealer penetration
into dentinal tubules, my recommendation is to accept with minor revision.
The information within this manuscript is accurate and should have minor corrections
on the grammar and language before it is accepted for publication. The study does
bring substantial contribution to the field of study. All information is well discussed,
and the results support conclusions.
1. My suggestion is to change the title to: Effect of different irrigation solutions on
the diffusion of MTA cement into the root canal dentin.
Answer: Thank you for your suggestion. The title was changed accordingly

Reviewer 3 Report

Authors focused on the in vitro analyzes of the penetration of a silicate root canal sealer into dentinal tubules after the use of different endodontic irrigating solutions.

 Manuscript has been prepared properly. Well-described methodology of the research is followed by the adequate discussion over the results of the analyses. In general, the paper is worth considering for publication in the Journal but some improvements are suggested. They all are provided in more detail below.

  • Abstract of the manuscript should be supplemented with the brief information concerning the novelty of the studies and their meaningless.
  • Additional paragraph concerning the currently used endodontic sealers needs to be added to the Introduction section (there is only one sentence in line 108). Next, the same section should be supplemented with brief information on the procedures currently used for the root canal obturation and what about penetration depth, what is already known and what is necessary to improve.
  • Section 2. Root canal filling.. contains the following sentence: “The root canal orifices were sealed using Cavit”. Please, explain why do you use Cavit instead of IRM capsules. In line 168..it is not clear did you use polishing under water cooling or not.
  • In results, CHX group obtained significant differences of sealer penetration between apical and middle section (it did not occur in NaOCl group). Characterize briefly the mentioned differences as well as their impact on the penetration depth of root canal obturation material. Is it  due to small sample? Please explain.
  • The properties and activity (i.e. the interactions causing the difference) of the chlorhexidine should be discussed in more detail by Authors. In line 245 you mentioned insufficient contact time-did you follow some protocol and use some proposed time for irrigation or not. How long it should be?
  • In Conclusions should be added some importance for clinical use.

Author Response

Authors focused on the in vitro analyzes of the penetration of a silicate root canal
sealer into dentinal tubules after the use of different endodontic irrigating solutions.
Manuscript has been prepared properly. Well-described methodology of the research
is followed by the adequate discussion over the results of the analyses. In general, the
paper is worth considering for publication in the Journal but some improvements are
suggested. They all are provided in more detail below.
1. Abstract of the manuscript should be supplemented with the brief information
concerning the novelty of the studies and their meaningless.
Answer: Thank you for your valuable comments. Accordingly to your suggestion,
the abstract was improved to better reflect the study novelty and meaningless.
2. Additional paragraph concerning the currently used endodontic sealers needs
to be added to the Introduction section (there is only one sentence in line 108).
Next, the same section should be supplemented with brief information on the
procedures currently used for the root canal obturation and what about
penetration depth, what is already known and what is necessary to improve.
Answer: Information about the currently use of endodontic sealers and the
techniques for root canal obturation were added to the introduction section in
lines 93-107.
3. Section 2. Root canal filling.. contains the following sentence: “The root canal
orifices were sealed using Cavit”. Please, explain why do you use Cavit instead
of IRM capsules . In line 168..it is not clear did you use polishing under water
cooling or not
Answer: As this study intended to assess the sealer penetration at dentinal
tubules and not coronal microleakage, Cavit was used because of the sealing
capacity, availability and ease of handling (Line 255)
Information regarding the use of water during the polishing procedures was
added at line 167.
4. In results, CHX group obtained significant differences of sealer penetration
between apical and middle section (it did not occur in NaOCl group).
Characterize briefly the mentioned differences as well as their impact on the
penetration depth of root canal obturation material. Is it due to small sample?
Please explain.
Answer: We believe the observed differences between the CHX and the NaOCl
groups are due to the different chemical actions of both irrigants, mainly
regarding the organic material removal. This is discussed at line 256 The text
regarding this section was improved to clarify it.
5. The properties and activity (i.e. the interactions causing the difference) of the
chlorhexidine should be discussed in more detail by Authors. In line 245 you
mentioned insufficient contact time-did you follow some protocol and use
some proposed time for irrigation or not. How long it should be?
Answer: The chlorhexidine irrigation time information (3 minutes) was added in
line 153.
More information about the properties and activity (i.e. the interactions causing
the difference) was added between line 255 and line 267.
6. In Conclusions should be added some importance for clinical use.
Answer: Information regarding the possible clinical implications of the present
results was added to the discussion section at lines 310 and 312: “This study
supports that final irrigation with NaOCl increases the calcium silicate sealer
penetration, which results in better sealing, and consequently the possibility of
a higher endodontic success rate.”

Reviewer 4 Report

Unfortunately, this study has several shortcomings.

  • Abstract: statistical analysis failed to show any significant differences between the groups. Thus, statements such as “obtained a higher mean percentage of sealer penetration” are not justified. The same is true for the final conclusions (page 9). Moreover, there is considerable confusion when looking at the results section. In the abstract it is stated that Kruskal-Wallis revealed no significant differences while in the results section significant differences are listed. Units are missing in all tables and figures. What is the meaning of “sealer penetration index” and how was it calculated?
  • MTA-Fillapex is not an appropriate sealer when trying to assess bioceramic-based sealers. It is just a conventional sealer with mixing in of MTA (see Assmann et al., J Endod 2015).
  • The exact levels of the transverse sections must be given
  • The root sections should be divided into four quadrants (buccal, mesial, oral, distal) due to the well-known butterfly-effect of the dentinal tubules. Accordingly, statistical analysis must be performed separately for all root levels and quadrants.
  • The use of rhodamine B is not a reliable method to assess sealer penetration. At best, the penetration of this dye into the dentinal tubules can be observed. Two studies using the calcium-affine marker Fluo-3 to investigate the penetration depth of calcium silicate-based sealers, reported impressively inferior penetration depths compared to values that can be expected from studies using rhodamine B (Jeong JW, DeGraft-Johnson A, Dorn SO, di Fiore PM. Dentinal Tubule Penetration of a Calcium Silicate–based Root Canal Sealer with Different Obturation Methods. J Endod 2017;43:633–7; Coronas VS, Villa N, Nascimento AL do, Duarte PHM, da Rosa RA, Só MVR. Dentinal tubule penetration of a calcium silicate-based root canal sealer using a specific calcium fluorophore. Braz Dent J 2020;31:109–15). Tis observation clearly points out that rhodamine B do not provide reliable results.

Due to these weaknesses this article cannot be recommended for publication.

Author Response

Unfortunately, this study has several shortcomings.
1. Abstract: statistical analysis failed to show any significant differences between
the groups. Thus, statements such as “obtained a higher mean percentage of
sealer penetration” are not justified. -
Answer: Statistical analysis of the obtained data showed that although no
differences were observed on the apical section, differences between the groups
were evident on the middle section of the samples. The abstract section was
modified to clarification.
2. The same is true for the final conclusions (page 9). Moreover, there is
considerable confusion when looking at the results section. In the abstract it is
stated that Kruskal-Wallis revealed no significant differences while in the
results section significant differences are listed.
Answer: The reviewer is correct. Corrections were made to the manuscript to
clarify that although no differences were observed on the apical section,
differences between the groups were evident on the middle section of the
samples.
3. Units are missing in all tables and figures. What is the meaning of “sealer
penetration index” and how was it calculated?
Answer: We appreciate the reviewer comment. The units were added to the
figures and tables. The sealer penetration index refers to the area occupied by
the sealer on the tooth wall. The sealer penetration in each section was
calculated by the following formula: sealer penetration area divided by the canal
circumferential area. This information was clarified at lines 186 and 187.
4. MTA-Fillapex is not an appropriate sealer when trying to assess bioceramicbased sealers. It is just a conventional sealer with mixing in of MTA (see
Assmann et al., J Endod 2015).
Answer: We agree with the reviewer. MTA-fillapex was chosen because is widely
used in the clinic with good results. As described in the introduction section
(line108), it is representative of an MTA-based sealer, both other components
are present.
5. The exact levels of the transverse sections must be given;
Answer: The information regarding the transverse sections was added at lines
171 and 172: “at 5 and 10 mm to the apex, obtaining dentin discs of the middle
and apical thirds of each root.”
6. The root sections should be divided into four quadrants (buccal, mesial, oral,
distal) due to the well-known butterfly-effect of the dentinal tubules.
Accordingly, statistical analysis must be performed separately for all root levels
and quadrants.
Answer: We agree that the root sections should be divided into 4 quadrants. We
will definitely include your point as a consideration for a future study. We have
now acknowledged this and suggested it as a topic for further research in the
discussion section. Thank you for the suggestion.
7. The use of rhodamine B is not a reliable method to assess sealer penetration.
At best, the penetration of this dye into the dentinal tubules can be observed.
Two studies using the calcium-affine marker Fluo-3 to investigate the
penetration depth of calcium silicate-based sealers, reported impressively
inferior penetration depths compared to values that can be expected from
studies using rhodamine B (Jeong JW, DeGraft-Johnson A, Dorn SO, di Fiore PM.
Dentinal Tubule Penetration of a Calcium Silicate–based Root Canal Sealer with
Different Obturation Methods. J Endod 2017;43:633–7; Coronas VS, Villa N,
Nascimento AL do, Duarte PHM, da Rosa RA, Só MVR. Dentinal tubule
penetration of a calcium silicate-based root canal sealer using a specific calcium
fluorophore. Braz Dent J 2020;31:109–15). Tis observation clearly points out
that rhodamine B do not provide reliable results.
Answer: We understand the reviewer comment that more precise methods can
be used to assess sealer penetration. However, rhodamine B use is widely
described as a methodology to determine the sealer penetration into dentinal
tubules.
Following the reviewer suggestion we added to the study limitations a statement
regarding the use of more specific techniques (line 296) and also referred to it at
the future directions paragraph (lines 301).
Due to these weaknesses this article cannot be recommended for publication.

Round 2

Reviewer 1 Report

Thank you for giving me this opportunity to re-review the revised manuscript. I carefully reviewed the submitted set of the revised manuscript and found it merits of publication.

Author Response

Thank you much for your suggestions to the manuscript.

Reviewer 4 Report

Although the manuscript has been extensively revised it still suffers from several flaws:

  • Abstract: „NaOCl promotes a more uniform sealer penetration, which can correlate with better sealing and, consequently, higher endodontic treatment success” > this is not a conclusion but mere a highly speculative statement. Neither “treatment success” nor “sealing” ability were evaluated in this study. Thus, this conclusion is misleading and not at all justified.
  • Throughout the entire manuscript: please replace “root filling” by “root canal filling”, as we do not obturate roots but canals.
  • A total of 29 teeth were used and divided into 2 experimental and one control group. Thus, the exact sample size for each group must be given.
  • The concern raised in the fist review (The root sections should be divided into four quadrants (buccal, mesial, oral, distal) due to the well-known butterfly-effect of the dentinal tubules. Accordingly, statistical analysis must be performed separately for all root levels and quadrants) has not been taken into consideration. Without analysis of the four quadrants the standard of this study is considerably below the quality of recently published articles.

Therefore, on the whole, the manuscript cannot be recommended for publication.

Author Response

1.“NaOCl promotes a more uniform sealer penetration, which can correlate with better sealing and, consequently, higher endodontic treatment success” > this is not a conclusion but mere a highly speculative statement. Neither “treatment success” nor “sealing” ability were evaluated in this study. Thus, this conclusion is misleading and not at all justified.

Answer: The abstract has been extensively revised, and the mentioned clinical implication was added at the request of one of the reviewers.

  1. Throughout the entire manuscript: please replace “root filling” by “root canal filling”, as we do not obturate roots but canals.

Answer: The document was verified, and the alteration was made throughout the entire manuscript.

  1. A total of 29 teeth were used and divided into 2 experimental and one control group. Thus, the exact sample size for each group must be given.

Answer: The exact sample size for each group was added in the abstract (line 37 and 38) and in the manuscript (line 149 to 152).

4.The concern raised in the fist review (The root sections should be divided into four quadrants (buccal, mesial, oral, distal) due to the well-known butterfly-effect of the dentinal tubules. Accordingly, statistical analysis must be performed separately for all root levels and quadrants) has not been taken into consideration. Without analysis of the four quadrants the standard of this study is considerably below the quality of recently published articles.

Answer: We agree that the root sections should be divided into 4 quadrants and your point was included as a consideration for a future study. We have now acknowledged this and suggested it as a topic for further research in the discussion section. However, our study design follows several studies published in the recent literature, in which the comparison was made between root thirds, not taking into consideration the quadrants.

  • Maybell Tedesco, Marcelo Chain, Wilson Felippe, Ana Alves, Lucas Garcia, Eduardo Bortoluzzi, Mabel Cordeiro, Cleonice Teixeira. Correlation between bond strength to dentin and sealers penetration by push-out test and CLSM analysis. Braz Dent J. 2019, 30, 555-562.
  • Coronas VS, Villa N, Nascimento AL do, Duarte PHM, da Rosa RA, Só MVR. Dentinal tubule penetration of a calcium silicate-based root canal sealer using a specific calcium Braz Dent J, 2020, 31, 109–15.
  • El Hachem, R.; Le Brun, G.; Le Jeune, B.; Pellen, F.; Khalil, I.; Abboud, M. Influence of the EndoActivator Irrigation System on Dentinal Tubule Penetration of a Novel Tricalcium Silicate-Based Sealer.  J.20186, 45.
  • De Bem, I.; Oliveira, R.; Weissheimer, T.; Bier, C., Só, M.; Rosa, R. Effect of ultrasonic activation of endodontic sealers on intratubular penetration and bond strength to root dentin. J Endod. 2020, 46, 1302-08.